# Investigation of the critical factors required to improve the disclosure and discussion of harm with affected women and families: a study protocol for a qualitative, realist study in NHS maternity services (the DISCERN study)

Mary Adams ![ORCID],[1] Rick Iedema ![ORCID],[2] Alexander Edward Heazell ![ORCID],[3] Maureen Treadwell,[4] Maria Booker,[5] Charlotte Bevan,[6] Julie Hartley,[1] Jane Sandall[1]

For numbered affiliations see end of article.

**Correspondence to**
Dr Mary Adams;
mary.adams@kcl.ac.uk

## ABSTRACT

Patients and families are entitled to an open disclosure and discussion of healthcare incidents affecting them. This reduces distress and contributes to learning for safety improvement. Complex barriers prevent effective disclosure and continue in the English NHS, despite a legal duty of candour. NHS maternity services are the focus of significant efforts to improve this. There is limited understanding of how, and to what effect, they are achieving this.

**Methods and analysis** A 27-month, three-phased realist evaluation identifying the critical factors contributing to improvements in the disclosure and discussion of incidents with affected families. The evaluation asks 'what works, for whom, in what circumstances, in why respects and why?'.

Phase 1: establish working hypotheses of key factors and outcomes of interventions improving disclosure and discussion, by realist literature review and in-depth realist interviews with key stakeholders (n=approximately 20]

Phase 2: refine or overturn hypotheses, by ethnographic case-study analysis using triangulated qualitative methods (non-participant observation, interviews (n=12) and documentary analysis) in up to 4 purposively sampled NHS trusts.

Phase 3: consider hypotheses and design outputs during seven interpretive forums.

**Ethics and dissemination** Phase 1 study approval by King's College London's Ethics Panel (BDMRESC 22033) and National Research Ethical Approval for Phases 2–3 (IRASID:262197) (CAG:20/CAG/0121) (REC:20/LO/1152). Study sponsorship by King's College London (HS&DR 17/99/85).

Findings to be disseminated through tailored management briefings; clinician and family guidance (written and video); lay summaries, academic papers, and report with outputs tailored to maximise academic and societal impact. Views of women/family groups are represented throughout.

---

**Strengths and limitations of this study**

► The first investigation of interventions supporting disclosure in NHS maternity care.
► Methodology includes multidisciplinary co-design throughout the research cycle, with wider applicability for health service research.
► A realist approach examines various contexts, mechanisms and outcomes of intervention success.
► Research does not examine disengagement among some maternity providers.

---

## INTRODUCTION

Besides patients' entitlement to an open disclosure of clinical incidents affecting them[1] when disclosure processes are conducted well, they can reduce the emotional harm to all involved,[2] provide valuable opportunities for learning for improvement[3 4] and may reduce public or private expenditures on investigatory processes and legal and financial claims.[5] In the longer term, disclosure practices may revise patient–clinician hierarchies and/or encourage public trust in healthcare.[6]

However, there is often disagreement or uncertainty over the principles, purpose and impact of disclosure practices. Their enactment often falls short of the expectations of harmed individuals,[6] professional or organisational guidelines[7] and regulatory directives.[8] In the UK, fears of professional investigation, loss of reputation and blaming behaviours are common deterrents to more open communication with patients. Inflexible organisational procedures and limited investment in the training and support of clinicians undertaking disclosure and discussions with

**BMJ**

patients can mean that the needs of patients are not properly heard or responded to.[9]

In the English NHS, each clinician's professional duty of honesty[7] co-exists with 'Being Open' policy guidance[10] that is now underpinned by a statutory duty of candour on health providers.[11] In NHS maternity services, an array of national and local safety improvement programmes is ongoing to reduce safety incidents and to improve the care of women/families when serious incidents happen. Since 2017, in England, there have been further efforts to improve incident investigation and women/family involvement in these processes.[12 13] One driver for these interventions is the reduction of the escalating costs of clinical negligence claims in NHS maternity care.[14] There is limited understanding of how, in what situations, to what effect and for whom these various improvement programmes are being implemented in maternity services and service areas, and their consequence for women/families, clinicians and organisations. Evidence is required to understand which improvements address these different groups' needs and interests when a harmful incident happens and how.

The primary aim is to develop actionable evidence to inform maternity providers of those factors that, in particular situations, will improve the disclosure and discussion of adverse incidents in NHS maternity care with affected women/families. The secondary aim is the co-production of outputs (support materials for women/families, clinicians and managers, and practical recommendations for organisations) to benefit these groups during these events. The overarching research question is 'what are the critical factors that can improve the incidence and quality of open disclosure in NHS maternity services?'. We will explore the actual or anticipated consequences of ongoing improvement programmes for different participants in particular contexts.

The study is co-designed and co-produced with three national patient and public involvement and engagement (PPIE) partners, including charities and associations representing women/families who have experienced harm in maternity care. These partners are core members of our co-investigator group (CIG). The work is also informed by a wider project advisory group (PAG) that includes eight individual women and families with direct experience of harm in maternity care. The PAG advises on study progress, outputs and dissemination.

## METHODS AND ANALYSIS
### Study design
The ongoing interventions in NHS maternity care comprise policy-led, complex interventions promoted in services in the expectation that incidence and/or quality of women/family involvement in the disclosure and discussion of adverse events will be improved. A realist, theory-driven evaluation approach informs study design. In health service evaluations, realism examines the interaction between a complex social intervention

or 'programme', the contexts of strategic and everyday implementation work, and the un/intended outcomes for different participants.[15] Realist evaluation asks, 'what works, for whom, in what circumstances, in what respects and why?'.[16] Therefore, the approach does not seek to establish the overall effect of an intervention but to identify the critical factors (in realist terms, the 'generative mechanisms') that are likely to produce expected and unexpected effects for different groups in particular circumstances.[17] The approach enables the analysis of the non-linear, contingent qualities of interventions, including the situated conditions of success and failure.

Realist evaluation follows a staged approach that moves from identifying initial working and interconnected hypotheses from secondary sources to the 'testing' (refinement or overturning) of these hypotheses using primary data to identify those contingencies that drive and direct improvement.[18]

The 3-phased, 27-month, qualitative study will:
► Establish a series of working hypotheses by literature scoping and expert stakeholder consultation (Study Phase 1a).
► 'Test' these working hypotheses by in-depth interviews with a range of national stakeholders (Study Phase 1b) and by ethnographic research (Study Phase 2).
► Agree with national stakeholders and different interest groups and experts, the critical factors required to trigger and sustain interventions to improve the disclosure and discussion of harm with women/families in NHS maternity care, and co-design outputs to support these generative factors (Study Phase 3).
► Follow the quality standards for realist evaluation.[19]

The study runs from May 2019 to April 2022 with 9 months COVID-related research interruption.

See online supplemental appendix 1 for the DISCERN study flow diagram.

### Methods for each study phase
#### Phase 1a
The PROSPERO registered realist review will identify and extract literature from electronic databases from social science, nursing, and medical and health service management. The purpose of the review is descriptive and will identify papers and documents of relevance to disclosure and discussion of harm in international maternity services. Initial scoping searches will identify keywords, synonyms and spelling variations. Searching will include free text and database-specific subject headings (such as MeSH terms), advanced Boolean truncation, 'explode' and other techniques. The database search will be supplemented by 'snowball' strategies, scanning reference lists from relevant papers and advice from subject experts. Inclusion criteria will be 'maternity and perinatal services' and 'disclosure and discussion of harm'. Following realist methodology for data gathering,[20] no pre-determined exclusion criteria concerning research methods will be applied. Grey literature, including policy reports, service guidance and public and professional commentary, will

be retrieved. The search is limited to English language documents. Identified full-text papers will be rescreened for both inclusion and exclusion criteria and relevance and rigour. If over 50 papers are identified, those most relevant to the emerging programme theories will be selected. A structured realist data extraction form will be developed to document item, context, intervention details/indications of mechanism/outcome relationships, factors affecting implementation and evidence of improvement programme outcomes.

Following RAMESES guidance[19], a realist analysis of extracted data will develop context–mechanism–outcome (CMO) configurations. From these, and with our stakeholders, we will identify initial (Phase 1a) programme theories and potential middle-range theories for examining the critical factors that influence improvements and their consequences for women/families, clinicians and services. One-to-one discussion in the CIG and consultation with the PAG, comprising academics, policy-makers, patient and public advocates and front-line clinicians, will develop and validate our secondary data analysis.

### Phase 1b
One-to-one, open-ended, audio-recorded telephone interviews conducted with approximately 20 national stakeholders in service improvement and patient/family engagement in NHS maternity care using a topic guide developed from Phase 1a findings and piloted with co-investigators. This interview study will (1) explore interviewees' various experiences and views concerning the programme theories developed from the realist review,[21] (2) gather 'soft intelligence' on site selection for Study Phase 2 and (3) establish stakeholders' views on how to maximise research impact[22] for Study Phase 3.

Interviewees will be sampled for maximum diversity across national stakeholder groups. Advice from the CIG and the PAG and snowball sampling with interviewees will guide the interviewee approach and selection.

See online supplemental appendix 2 for the flow diagram of study recruitment and data management for Study Phase 1b.

### Phase 2
Up to four NHS maternity services, identified as high performing or improving (in terms of supporting interventions for strengthening disclosure practices with women/families after harm in maternity care), will be selected as ethnographic case studies. An ethnographic methodology will enable in-depth interpretation of the context and practices of clinical or organisational relationships, including the situated and often taken-for-granted qualities of these practices, including their negotiation and legitimation.[23] This approach is important for understanding the local reception, adjustments and viability of new interventions.[24] Ethnographic case studies enable detailed data to be analysed dynamically regarding particular organisational and service contexts, events and activities, and as a series of comparative case studies.

A case study comprises 10×3-day visits, completed over 10 months, with periods between visits for researchers to discuss and compare emergent findings. Initial hypotheses and their refinements or revisions (developed in Study Phase 1) will focus the comparative case studies. However, they will also document (1) how relevant policy and procedures are promoted, managed and monitored as well as interpreted and evaluated by recipients (along with the challenges and opportunities to this), (2) factors that influence the 'when and how' of procedural enactments in actual clinical settings by different participants, including women/families, clinicians, administrators and management and governance teams, (3) the interplay of formal and informal local workplace cultures and wider public and policy debates and (4) personal, team and service histories of disclosure experiences, including their potential to influence wider revisions in the 'Being Open' agenda, patient–clinician–manager relationships and their acceptance as routine.

Our sampling follows social science principles. Due to expected challenges in engaging managers or clinicians in some maternity services, the sampling frame includes only cases of positive deviance to maximise engagement in the study. A positive deviance approach posits that behaviours (and situations) the produce poor practice or experience are variations on the same processes that produce successful practices. Therefore, focusing on success is a more effective tactic for understanding how improvements are achieved.[25] Sampling includes maximum diversity between services so that opportunities to learn about in-case and cross-case variation are maximised.[26] A shortlisting of potential case study sites will use public data set scores and information on service provision (size, population served and neonatal care provision).

Further information on the sampling framework is described in online supplemental appendix 3.

Approach and recruitment of selected organisations, site principal investigators (PIs) and key gatekeepers are tailored from a theory-based, business-model approach to recruitment to clinical studies. A description of the application of this business model to the study is described in online supplemental appendix 4.

Each ethnographic case study will triangulate findings collected by the following methods:

Focused observations [estimated 25 hours] of formal and informal events and routines intended to secure improvement in disclosure of adverse incidents with women/families (eg, service-level and Trust-level meetings on serious incident reporting and analysis; training activities; promotion and publicity work; and meetings between patient liaison and clinical teams. Observations may include meetings between family members and organisational representatives.

In-depth return interviews with approximately 12 staff [some at two time-points] involved with, or affected by, the improvement work (eg, clinical and

governance leads; patient support personnel; more junior clinicians).

Documentary data including redacted incident reports; meeting minutes, action plans; organisational and service bulletins; training reports; quality and safety improvement reports; and incident reports from families to construct a retrospective picture of open disclosure work within that case study.

With participants' informed consent, in-depth interviews will be audio-recorded on encrypted devices and transcribed verbatim. The observational data will be recorded as low inference notes and later word-processed by that researcher. All documentary data will be redacted before inclusion in the study.

### Phase 3

Following initial analysis of case-study data, seven forums of purposively selected individuals will validate findings[27] and forum participans will co-design, with the CIG, training materials from the study findings. Audio-recorded forum discussions will be thematically analysed to (1) further validate and develop findings on the critical factors and situations that support disclosure processes and (2) support output co-development.

Potential participants will be identified during participation with Phase 1 or Phase 2 and sampled for maximum variation in their experiences of disclosure practice improvements. A maximum of 25 participants will be invited to any one of the 7 local forums. A final national forum, including local forum representatives and national policy-makers, will accommodate up to 60 participants and be externally facilitated. The national event will be hosted at the sponsoring university. Wider consultations of study findings and output development will be with established online forums for harmed patients/families and clinicians interested in maternity safety improvement.

### Public and patient participation and engagement

To ensure that the views and interests of women/families, and the wider public, remain the focus of the study, our PPIE strategy supports representation at all study phases (study design, management, data interpretation, co-development of outputs, reporting of findings and dissemination of research findings, along with other outputs). We have followed the new national standards,[28] which provide a series of indicators to support structures and behaviours that support PPIE in study governance and study impact. The approach reflects current policy emphasis on participatory approaches to healthcare service quality and safety improvement. This is to be achieved by:

▶ The involvement of women/families and their charity and association representatives in all stages of the research cycle (study application, research planning, research activity, study steering, write-up and dissemination of findings).

▶ A multidisciplinary and multi-sector research team that includes public members.
▶ Ongoing collaboration with an extensive and varied PAG, including their involvement in the summative forum.
▶ Use of interpretive forums (described above).

The study recognises the need for co-development of outputs so that different interests, values and subject positions can be accommodated. Standards for reporting qualitative research[29] have been followed in the protocol design.

### DATA ANALYSIS

Data will be organised and analysed following a realist logic approach. Retroductive analysis (moving between inductive and deductive approaches) will identify and explore main context–mechanism–outcome (CMO) patterns, particularly the generative causal mechanisms shaping these configurations. This approach is to refine, develop or refute initial programme theories developed progressively from Study Phase 1 (realist literature review) and Study Phase 2 (ethnographic case studies). Additional CMO configurations may be elucidated at Study Phase 2 and examined retroductively across this data set to inform, revise or add to the programme theories. Data analysis will be conducted at the close of each study phase to maintain an ongoing and iterative approach to programme theory refinement. Two researchers trained in realist evaluation will independently identify and explore main CMO configurations. Then with additional researchers and stakeholders, explore programme theory refinement in the light of these. Phase 2 interview, ethnographic and documentary data, will also be analysed in a triangulating fashion to develop concrete descriptions of the generative mechanisms underlying open disclosure interventions and the contextual factors that trigger effects at particular times. We will seek to identify convergent and contradictory evidence within and then across the interventions' different organisational, interpersonal and individual contexts. Coding and data management will be organised and audited using qualitative data analysis software, Nvivo (V.12), and CMO data coding sheets will be developed for the analysis.

Where possible, cross-tabulations and scattergrams will be used to plot the relationship between apparently anomalous aspects of improvement work in this area and other social variables (eg, distinctive service or organisational histories, resourcing agreements or public involvement structures and relationships).

A stakeholder analysis using the RAPID AIIM matrix[22] may be used to assist in mapping participant interests in the subject area and their influence over local and national improvement work.

The hypotheses generated will also be developed using conceptual frameworks drawn from the social sciences (eg, normalisation process theory, an institutional logics approach and approaches to trust in healthcare

organisations) and perspectives on team and organisational safety and resilience. RAMESES II reporting standards for realist evaluations will be followed for the presentation of data collection, findings and discussion.[30]

## ETHICS AND DISSEMINATION

Outputs will be robust and actionable evidence about the organisational strategies. These team, service and network practices can support and enhance the quality and extent of open disclosure within NHS maternity care and wider NHS care provision within particular circumstances and with respect to different interests groups. Our findings are intended to impact three organisational areas of healthcare: the micro-level of disclosure processes and relationships; meso-level of service management and organisation of these processes; and macro-level of policy. Therefore, our dissemination strategy, with outputs tailored to the audience,[31] includes:

► Co-designed briefing documents and guidance for maternity service managers and policy-makers.
► Co-designed information materials (likely, short guidance sheets) and video animations for services, clinicians and women/families (and promotion of these by our Stage 3 study participants).
► Accessible public information and lay summaries of study findings disseminated through our dedicated Twitter account and website.
► Peer-reviewed academic journal articles (including the final NIHR study report).

A summary of findings was sent to all study participants who have consented to receive this information.

More widely, the dissemination plan is underpinned by all RCUK identified factors that help to generate academic and societal impact. The research will enhance the evidence base for improving the use and quality of open disclosure in NHS maternity care, influencing policy and practice; will produce theoretically informed knowledge of the critical factors for fostering, enhancing and sustaining open disclosure; and will support the growth of networks and knowledge exchange between NHS maternity providers, women/family interest groups and associations, policy-makers and non-maternity NHS providers.

### Ethical considerations

We have identified this study as a high risk, qualitative research study because we are:

► Conducting research on a highly sensitive and potentially emotive topic for all concerned.
► Interviewing and observing clinicians and managers whose care might have resulted in harm.
► Observing meetings with women/families and managers that may become adversarial.

The primary ethical issues requiring mitigation are:

► Non-consent or withdrawal of consent to study participation (eg, by some staff during research observation of staff meetings and routines): in these situations, we will always respect the withdrawal of study participation

and state this in all participant information and consent forms. We also will seek to focus observations of collective routines on more public workplace activities (eg, training events).
► Disclosure of negligence (eg, during staff and women/family representative interviews): we will make it clear in all participant observations that we have a duty to disclose staff negligence. We will establish a system for researchers concerned about negligence to alert the PI within an established time frame and prompt notification of the relevant Freedom to Speak Up Guardian.
► Participant engagement (by trusts, services, units and teams who feel judged by the study sampling procedures and the study topic): we know that obtaining initial and ongoing engagement with participants is possible if done sensitively and ethically. We will reassure all trusts and services that we will not be approaching, interviewing or observing women and families directly.
► Non-malfeasance regarding study participants: in research information and methods, we will follow current guidance on interviewing people about potentially sensitive topics or traumatic experience[32 33] and specialist (one-to-one) advice from our PAG.
► Non-malfeasance regarding PPIE representatives (study co-applicants, PAG members and forum participants): we have structured PPIE representation across the study, so that leadership and support roles are clearly defined. We have ensured that all PPIE co-applicants and PAG members are highly experienced in research and engaging with diverse interest groups. We have also ensured linkage between the PPIE PAG lead and the study PAG lead to ensure consistency of support and representation.

Issues of anonymity, confidentiality and informed consent will be addressed in the recruitment of all participants, data collection processes and data storage. The principles of beneficence and non-malfeasance will be adhered to.

Throughout the study, the MRC/HRA guidance 'GDPR and Data Protection Act 2018: key facts for researchers' has been followed.

### Study strengths and limitations

► This is the first study to investigate the implementation of interventions for disclosure and discussion of harm in NHS maternity care.
► Project methodology includes multidisciplinary and user representative co-design of study questions, research focus, study outputs and dissemination strategy, with wider applicability in health service research.
► A realist approach identifies 'key ingredients for potential success' for maternity providers engaged in disclosure improvement.
► The study does not examine reasons for disinterest or disengagement among some maternity providers.

**Author affiliations**
¹Faculty of Life Science and Medicine, Department of Women and Children's Health, King's College London, London, UK
²Centre for Team Based Practice and Learning in Health Care, King's College London, London, UK
³Division of Developmental Biology and Medicine, School of Biological Sciences, The University of Manchester, Manchester, UK
⁴Birth Trauma Association, Derbyshire, UK
⁵Birthrights, London, UK
⁶Stillbirth and Neonatal Death Charity, London, UK

**Contributors** MA drafted and revised the protocol with ongoing contributions from JS, RI, AEH, MT, MB and CB. JS assisted with proofreading the protocol for submission for publication.

**Funding** This work is supported by the National Institute of Health Research (NIHR), Health Service and Delivery Stream (grant number: HS&DR 17/99/85). The study has been adopted by the NIHR Applied Research Collaboration, South London.

**Competing interests** None declared.

**Patient consent for publication** Not applicable.

**Provenance and peer review** Not commissioned; externally peer reviewed.

**ORCID iDs**
Mary Adams http://orcid.org/0000-0003-1276-617X
Rick Iedema http://orcid.org/0000-0001-6792-1048
Alexander Edward Heazell http://orcid.org/0000-0002-4303-7845

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
