## [Reviewer comments · BMJ Open]

ARTICLE DETAILS

TITLE (PROVISIONAL)	INVESTIGATION OF THE CRITICAL FACTORS REQUIRED TO IMPROVE THE DISCLOSURE AND DISCUSSION OF HARM WITH AFFECTED WOMEN AND FAMILIES: STUDY PROTOCOL FOR A QUALITATIVE, REALIST STUDY IN NHS MATERNITY SERVICES [THE DISCERN STUDY].
AUTHORS	Adams, Mary; Iedema, Rick; Heazell, Alexander; Treadwell, Maureen; Booker, Maria; Bevan, Charlotte; Hartley, Julie; Sandall, Jane

VERSION 1 – REVIEW

REVIEWER	Geoff Wong Oxford University, Nuffield Department of Primary Care Health Sciences
REVIEW RETURNED	30-Oct-2021

GENERAL COMMENTS	Thank you for asking me to review this manuscript. I am not a content expert and so have not made any comments on the topic content of this manuscript. As my area of expertise is realist research approaches (namely realist review/synthesis and realist evaluation) I have focused almost exclusively on providing comments related to these two approaches. Overall, it makes sense to use a realist review followed by a realist evaluation approach to make sense of this topic area. The rationale for the use of such approaches is provided and reasonable. The research questions are amenable to being answered using the realist approaches proposed. The details provided for the realist review's protocol is rather brief, but sufficient. The methods planned for primary data collection make sense and should enable the authors to get sufficient relevant data for their realist evaluation. My major concern in this protocol is with the planned data analysis approach used for the realist review and realist evaluation. My concerns are detailed below: - Page 6 of 14 lines 14 to 15: It is not appropriate to undertake only a thematic analysis of the data in a realist review. This is not consistent with the RAMESES quality standards for realist syntheses (see Item 2 of: https://www.ramesesproject.org/media/RS_qual_standards_researchers.pdf). A realist logic of analysis must be used to analyse the data and a programme theory (of some sort) must be developed (see item 4 of
--

	link above). - Page 7 of 14 line 15 and Data analysis section (lines 23 to 40): If the authors are planning to do a realist project (as suggested in their title), then my assumption would be that Phase 2 (and possible 3) uses the realist evaluation approach. If this is the case then I have significant concerns about the planned approach to data analysis in Phases 2 and 3. As for Phase 1 a realist logic of analysis should be used to develop the context-mechanism-outcome-configurations that underpin the programme theory. In a realist evaluation developing a programme theory (item 3) and applying a realist logic of analysis item 2) are expected - see: https://www.ramesesproject.org/media/RE_Quality_Standards_for_evaluators_and_peer_reviewers.pdf. Only doing thematic analyses is both inadequate and inappropriate in a realist evaluation (and review). I do have some minor concerns that I would be grateful if the authors would kindly address. - Page 6 of 14 lines 6 to 8: I would suggest that you do develop some inclusion and exclusion criteria as otherwise it would be challenging to whittle down the hits from the searches. These inclusion and exclusion criteria can be quite broad - e.g. any type of document, any outcomes, disclosure and discussion of harm and maternity services. On a minor point the project is called RAMESES. - Page 6 of 14 line 10: Practically it is hard to screen the abstracts of identified papers against the criteria of relevance and rigour. It is easier to do this once you have sifted the documents by title / abstract and then full text against the inclusion and exclusion criteria. My advice would be to sift documents in a three staged process: 1) by title and abstract against inclusion and exclusion criteria. 2) full text against inclusion and exclusion criteria. 1) and 2) will then provide a virtual pile of paper that is more likely to contain relevant data. You can then use relevance and rigour to sift through this pile to make final decisions on inclusion. In terms of the operationalisation of rigour, I would suggest the authors read the following book chapter to see if it helps them decide on what to do. Data gathering for realist reviews: Looking for needles in haystacks. Wong G. In: Emmel N, Greenhalgh J, Manzano A, Monaghan M, Dalkin S, editors. Doing Realist Research. London: Sage, 2018. - Page 6 of 14 line 21: Please clarify if your "initial hypotheses" is the same thing as your initial programme theory or is it the programme theory that has been developed and 'tested' (confirmed refuted or refined) from the realist review? - Page 6 of 14 line 38: Please clarify if your "initial hypotheses" is the same thing as your initial programme theory or is it the programme theory that has been developed and 'tested' (confirmed refuted or refined) from the realist
--	--

	review? - Page 8 of 14 line 44: If the authors are indeed planning to use a realist evaluation approach in Phase 2, then they should follow the: - quality standards for realist evaluations when designing their evaluation: https://www.ramesesproject.org/media/RE_Quality_Standards_for_evaluators_and_peer_reviewers.pdf - reporting standards for realist evaluations when reporting their evaluation: https://bmcmedicine.biomedcentral.com/articles/10.1186/s12916-016-0643-1. Good luck with your project.
--	---

VERSION 1 – AUTHOR RESPONSE

	Reviewers Comment	Authors' Response and Corrections Made
		Thank you for reviewing our protocol so thoroughly. We feel very privileged to have an international realist expert look over this and suggest revisions. This has helped us to rethink our approach to data analysis at all study stages. Since submission of this protocol for publication we have completed realist training and jointed a realist study group that has helped our practical understanding and ongoing appreciation of this approach.
1	- Page 6 of 14 lines 14 to 15: It is not appropriate to undertake only a thematic analysis of the data in a realist review. This is not consistent with the RAMESES quality standards for realist syntheses (see Item 2 of: https://www.ramesesproject.org/media/RS_qual_standards_researchers.pdf). A realist logic of analysis must be used to analyse the data and a programme theory (of some sort) must be developed (see item 4 of link above).	Since submission of this protocol we have progressed this Stage 1 work and we did not conduct a thematic analysis of the literature. Instead, we used a logic of realist analysis approach that sought to elicit a series of underlying assumptions about how the interventions described are expected to work and organised this data extraction as a series of if/then hypotheses. Therefore, we have deleted 'thematic analysis of extracted data' and replaced this this: Following RAMESES guidance¹⁹, a realist analysis of extracted data will develop Context-Mechanism-Outcome configurations. (Page 4 paragraph 5)
2	Page 7 of 14 line 15. If the authors are planning to do a realist project (as suggested in their title), then my assumption would be that Phase 2 (and possible 3) uses the realist	Response to suggested revisions for data analysis Study Phase 3. Thank you for this advice. We regret that we lack the capacity to complete a realist evaluation approach for Phase 3. The

evaluation approach. If this is the case then I have significant concerns about the planned approach to data analysis in Phases 2 and 3.	purpose of this approach is 'to reach agreement on our findings with stakeholders and co-design outputs that will support generative factors' (Page 3 of 16, line 56-8). We feel that a realist evaluation of our own interpretation and dissemination planning activities would constitute another and distinctive research project. Therefore we would like to keep the statement that "Audio-recorded forum discussions will be thematically analysed to (a) further validate and develop findings on the critical factors and situations that support disclosure processes; (b) support output co-development." (Page 6 paragraph 6). However, we appreciate that two statements in the protocol indicate that we are conducting a realist analysis of this stage. Therefore we have made the following revisions  1. In the Abstract we have deleted "Phase 3: refine hypotheses' and we have replaced it with "Phase 3: consider hypotheses' (as we are unable to made any further realist refinements at this stage). 2. In Methods for Each Study Phase: Phase 3 (page 5, paragraph 4), we have deleted 'contribute to the interpretation' (so that our validation of findings rather than any further realist hypothesis development)
3 . Data analysis section (lines 23 to 40): If the authors are planning to do a realist project (as suggested in their title), then my assumption would be that Phase 2.... uses the realist evaluation approach.If this is the case then I have significant concerns about the planned approach to data analysis in Phases 2 and 3. As for Phase 1 a realist logic of analysis should be used to develop the context-mechanism-outcome-configurations that underpin the programme theory. In a realist evaluation developing a programme theory	Response to suggested revisions for data analysis Study Phase 2. Thank you for this observation of our description of data analysis. We are certainly planning to conduct a realist analysis at Phase 2 of the study. We are organising our data analysis using a realist logic, which seeks to identify generative causation, identifying mechanisms in relation to context and outcomes. Therefore, we delete the following sentences (Page 5 of 16, lines 25-31) "Data will be analysed iteratively, with reading and re-reading of transcripts in refine or overturn initial hypotheses^{xxviii}. Some data analysis will be conducted at the close of each Study Phase, to maintain a staged approach to hypotheses

(item 3) and applying a realist logic of analysis item 2) are expected - see: https://www.ramesesproject.org/media/RE_Quality_Standards_for_evaluators_and_peer_reviewers.pdf. Only doing thematic analyses is both inadequate and inappropriate in a realist evaluation (and review).	refinements. Following familiarisation with the anonymised transcripts, two experienced researchers will independently identify and explore emergent themes in relation to our initial hypotheses of the critical factors that support OD improvements, to what effect and for whom (as identified in Phase 1a). Phase 2 data will also be analysed in a triangulating fashion to develop concrete descriptions of the different factors, contexts and contingencies that foster and shape OD interventions and their various outcomes and effects in particular services. In-case analysis and cross-case comparisons, will help to refine our understanding of which critical factors may be fostered or sustained in different contexts and contingencies and produce different effects and outcomes^{xix}. We will seek to identify convergent and contradictory evidence within and then across the cases. Coding and data management will be organised and audited using qualitative data analysis software, Nvivo (V12). Two more senior researchers will check and help to refine data analysis. And replace them with: Data will be organised and analysed following a realist logic of analysis. Retroductive analysis (moving between inductive and deductive approaches) will identify and explore main Context-Mechanism-Outcome patterns, particularly the generative causal mechanisms shaping these configurations. This approach is to refine, develop or refute initial program theories developed progressively from Study Phase 1 (realist literature review) and Study Phase 2 (ethnographic case studies). Additional C-M-O configurations may be elucidated at Study Phase 2 and examined retroductively across this data set to inform, revise or add the program theories. Data analysis will be conducted at the close of each study phase to maintain an ongoing and iterative approach to programme theory refinement. Two researchers trained in realist evaluation will independently identify and explore main C-M-O configurations. Then with additional researchers and stakeholders, explore programme theory refinement in the light of these. Phase 2 interview, ethnographic and documentary data will also be analysed in a triangulating fashion to develop concrete descriptions of the generative mechanisms underlying OD interventions and the contextual factors that trigger effects at particular times. We will seek to identify convergent and contradictory evidence within and then across the interventions' different organisational, interpersonal and individual contexts. Coding and data management will be organised and audited using qualitative data analysis software, Nvivo (V12), and CMO data
---	---

		coding sheets will be developed for the analysis. (page 6 paragraph 9 to page 7 paragraph 1).
	Minor Concerns	
4	- Page 6 of 14 lines 6 to 8: I would suggest that you do develop some inclusion and exclusion criteria as otherwise it would be challenging to whittle down the hits from the searches. These inclusion and exclusion criteria can be quite broad - e.g. any type of document, any outcomes, disclosure and discussion of harm and maternity services.	This is a very helpful suggestion. We have now developed these criteria and revised the data gathering process, as advised. Therefore, we have deleted: “Following realist methodology, no specific pre-determined inclusion and exclusion criteria will be used to identify inclusions, however, we will report on areas of study design and evidence weakness and rank inclusions, following RAMSES project guidance. Grey literature, including policy reports, service guidance and public and professional commentary will be retrieved. The search is limited to English language documents. Abstracts of all identified papers will be screened for relevance and rigour. If over 50 papers are identified those most relevant to study aims and objectives will be selected. A structured realist data extraction form will be developed to document item; context; intervention details/indications of mechanism/outcome relationships; factors affecting implementation; evidence of improvement program outcomes. Thematic analysis of extracted data to develop early hypotheses of critical factors that influence improvements and their consequences for women/families, clinicians and services. One-to-one discussion in the CIG, and consultation with the PAG, comprising academics, policy-makers, patient and public advocates and front-line clinicians, to validate secondary data analysis” (page 4 parag. 1).

		and replaced by Inclusion criteria will be 'maternity and perinatal services' and 'disclosure and discussion of harm'. Following realist methodology for data gathering²⁰, no pre-determined exclusion criteria concerning research methods will be applied. Grey literature, including policy reports, service guidance and public and professional commentary, will be retrieved. The search is limited to English language documents. Identified full-text papers will be rescreened for both inclusion and exclusion criteria and relevance and rigour. If over 50 papers are identified, those most relevant to the emerging programme theories will be selected. A structured realist data extraction form will be developed to document item; context; intervention details/indications of mechanism/outcome relationships; factors affecting implementation; evidence of improvement program outcomes. Following RAMESES guidance¹⁹, a realist analysis of extracted data will develop Context-Mechanism-Outcome configurations. From these, and with our stakeholders, we will identify initial (Phase 1a) programme theories and potential middle-range theories for examining the critical factors that influence improvements and their consequences for women/families, clinicians and services. One-to-one discussion in the CIG and consultation with the PAG, comprising academics, policymakers, patient and public advocates and front-line clinicians, will develop and validate our secondary data analysis. (page 4 para 4-5 and page 5 para 1) We have referenced your 2018 paper on data gathering in realist reviews and the RAMESES guidance for qualitative research (see also item 6 below).
5	On a minor point the project is called RAMESES.	Our apologies for this. This spelling error has now been corrected in the revised. (page 4 para 5)
6	- Page 6 of 14 line 10: Practically it is hard to screen the abstracts of identified papers against the criteria of relevance and rigour. It is easier to do this	Many thanks for this advice. For study Phase 1, we did find that we had to revise this approach, as you suggest, and rescreen full-text

once you have a sifted the documents by title / abstract and then full text against the inclusion and exclusion criteria. My advice would be to sift documents in a three staged process: 1) by title and abstract against inclusion and exclusion criteria. 2) full text against inclusion and exclusion criteria. 1) and 2) will then provide a virtual pile of paper that is more likely to contain relevant data. You can then use relevance and rigour to sift through this pile to make final decisions on inclusion. In terms of the operationalisation of rigour, I would suggest the authors read the following book chapter to see if it helps them decide on what to do. Data gathering for realist reviews: Looking for needles in haystacks. Wong G. In: Emmel N, Greenhalgh J, Manzano A, Monaghan M, Dalkin S, editors. Doing Realist Research. London: Sage, 2018.	for first, inclusion and exclusion criteria and second, for ranking for relevance and rigour. This chapter has been very useful and it is now cited in the protocol here as it helped towards making practical decisions for management of the information. We have included your reference for the revised paragraph (see comment 6 reply above).
7 - Page 6 of 14 line 21: Please clarify if your "initial hypotheses" is the same thing as your initial programme theory or is it the programme theory that has been developed and 'tested' (confirmed refuted or refined) from the realist review?	Our clarification is as follows: We have deleted the term 'early hypotheses' and replaced it by From these, and with our stakeholders, we will identify initial (Phase 1a) programme theories and potential middle-range theories for examining the critical factors that influence improvements and their consequences for women/families, clinicians and services. (page 4 parag 5) The reason for adjusting this terminology is that we are using the programme theories developed from the realist review of the literature to 'test' empirically (confirm, overturn or develop) during our Phase 2 realist data collection and analysis.

		When our protocol was accepted by our funders and sponsors they requested some translations from realist terminology. We therefore redrafted the term 'initial programme theory' to 'initial hypothesis' (the set of assumptions often implicit in intervention design or enactment).
8	- Page 6 of 14 line 38: Please clarify if your "initial hypotheses" is the same thing as your initial programme theory or is it the programme theory that has been developed and 'tested' (confirmed refuted or refined) from the realist review?	We have revised the term 'early hypotheses' to This interview study will (i) explore interviewees' various experiences and views concerning the programme theory developed from the realist review²¹; (page 5 para 1) We are aware that there may be slippage between 'programme theories' and middle-range theory in terms of local assumptions over why an interventions might work and theoretical propositions on how it works (although, of course, the level of abstraction of the formal, middle-range theory will be very different). Therefore, we have intentionally left the possible connections that we will have not tightly defined the relationship between programme theories and the middle-range theories in this protocol.
8	- Page 8 of 14 line 44: If the authors are indeed planning to use a realist evaluation approach in Phase 2, then they should follow the: - quality standards for realist evaluations when designing their evaluation: https://www.ramesesproject.org/media/RE-Quality_Standards_for_evaluators_and_peer_reviewers.pdf	Thank you for this advice and reference. We will be following these during the study. Therefore, we have inserted the following references into the document: 1. In the final line of the Study Design section we have added:  • The study will follow the quality standards for realist evaluation¹⁹.

- reporting standards for realist evaluations when reporting their evaluation: https://bmcmedicine.biomedcentral.com/articles/10.1186/s12916-016-0643-1.	(page 4 para 3) We have also referenced: https://www.ramesesproject.org/media/RE_Quality_Standards_for_evaluators_and_peer_reviewers.pdf 2.In the final line of the Date Analysis section we have added: RAMESES II reporting standards for realist evaluations will be followed for the presentation of data collection, findings and discussion²⁸ (page 7, para 4) and referenced https://www.ramesesproject.org/media/RS_qual_standards_researchers.pdf).
---	---